# Genomic Insights of Wheat Root-Associated *Lysinibacillus fusiformis* Reveal Its Related Functional Traits for Bioremediation of Soil Contaminated with Petroleum Products

**DOI:** 10.3390/microorganisms12112377

**Published:** 2024-11-20

**Authors:** Roderic Gilles Claret Diabankana, Akerke Altaikyzy Zhamalbekova, Aigerim Erbolkyzy Shakirova, Valeriia Igorevna Vasiuk, Maria Nikolaevna Filimonova, Shamil Zavdatovich Validov, Radik Ilyasovich Safin, Daniel Mawuena Afordanyi

**Affiliations:** 1Laboratory of Molecular Genetics and Microbiology Methods, Kazan Scientific Center, Russian Academy of Sciences, 420111 Kazan, Russia; vivalalastochka@gmail.com (V.I.V.); sh.validov@knc.ru (S.Z.V.); 2Kazakh Scientific Research Institute of Plant Protection and Quarantine Named After Zh. Zhiembayev, Timiryazev 45, Almaty 050040, Kazakhstan; akzhamalbekova@niizkr.kz (A.A.Z.); ashakirova@niizkr.kz (A.E.S.); 3Academic and Research Centre, Institute of Fundamental Medicine and Biology, Kazan Federal University, 420008 Kazan, Russia; maria.filimonova@kpfu.ru; 4Centre of Agroecological Research, Kazan State Agrarian University, 420015 Kazan, Russia; dean.agro@kazgau.com; 5Tatar Research Institute of Agriculture, Kazan Scientific Center, Russian Academy of Sciences, 420111 Kazan, Russia

**Keywords:** azo dye, crude oil, phenol, polycyclic aromatic hydrocarbons, pangenome, xenobiotic assimilation

## Abstract

The negative ecological impact of industrialization, which involves the use of petroleum products and dyes in the environment, has prompted research into effective, sustainable, and economically beneficial green technologies. For green remediation primarily based on active microbial metabolites, these microbes are typically from relevant sources. Active microbial metabolite production and genetic systems involved in xenobiotic degradation provide these microbes with the advantage of survival and proliferation in polluted ecological niches. In this study, we evaluated the ability of wheat root-associated *L. fusiformis* MGMM7 to degrade xenobiotic contaminants such as crude oil, phenol, and azo dyes. We sequenced the whole genome of MGMM7 and provided insights into the genomic structure of related strains isolated from contaminated sources. The results revealed that influenced by its isolation source, *L. fusiformis* MGMM7 demonstrated remediation and plant growth-promoting abilities in soil polluted with crude oil. *Lysinibacillus fusiformis* MGMM7 degraded up to 44.55 ± 5.47% crude oil and reduced its toxicity in contaminated soil experiments with garden cress (*Lepidium sativum* L.). Additionally, *L. fusiformis* MGMM7 demonstrated a significant ability to degrade Congo Red azo dye (200 mg/L), reducing its concentration by over 60% under both static and shaking cultivation conditions. However, the highest degradation efficiency was observed under shaking conditions. Genomic comparison among *L. fusiformis* strains revealed almost identical genomic profiles associated with xenobiotic assimilation. Genomic relatedness using Average Nucleotide Identity (ANI) and digital DNA–DNA hybridization (DDH) revealed that MGMM7 is distantly related to TZA38, Cu-15, and HJ.T1. Furthermore, subsystem distribution and pangenome analysis emphasized the distinctive features of MGMM7, including functional genes in its chromosome and plasmid, as well as the presence of unique genes involved in PAH assimilation, such as *phn*C/T/E, which is involved in phosphonate biodegradation, and *nem*A, which is involved in benzoate degradation and reductive degradation of N-ethylmaleimide. These findings highlight the potential properties of petroleum-degrading microorganisms isolated from non-contaminated rhizospheres and offer genomic insights into their functional diversity for xenobiotic remediation.

## 1. Introduction

The contamination of soil with toxic heavy metals, petroleum, and byproducts, as well as azo dyes, poses a serious threat to living beings and ecosystems. Azo dyes, which are commonly used as synthetic dyes in the textile industry, and petroleum are particularly problematic pollutants [1,2,3]. Improper management of these environmental pollutants has led to the accumulation of harmful constituents that negatively impact and contribute to human disease and climate change [4,5,6]. Moreover, these hazardous substances directly lead to (bring about) several complex socio-economic consequences, including a decrease in agricultural and aquaculture productivity, leading to serious economic losses [7,8]. Several remediation techniques, including soil washing, in situ chemical oxidation, and thermal desorption, have been developed to process persistent pollutants from environmental matrices, with efficiency varying based on the characteristics of the contaminants [9]. Chemical remediation techniques, such as chemical oxidation, treat a range of contaminants but have limitations such as excessive costs and environmental feasibility [10]. Bioremediation as an eco-friendly management tool to manage pollutants tends to be the best alternative treatment process due to its cost-effectiveness and ability to naturally degrade pollutants into less toxic substances. Several studies have highlighted the ability of different microorganisms to degrade different contaminants, such as alkanes and polycyclic aromatic hydrocarbons (PAHs) present in petroleum, into carbon dioxide and water using a wide range of enzymes [11]. For example, *Pseudomonas* and *Rhodococcus* are known for hydrocarbon assimilating ability, while *Bacillus*, *Alcaligenes*, and *Enterobacter* assimilate various pollutants, including decomposition of azo dyes and heavy metals removal [12,13,14]. The ability of these microorganisms to degrade petroleum components is related to the presence of key genes in their genomic or plasmid DNA. Microbial key genes involved in the remediation of petroleum oil, such as alkane hydroxylase and monooxygenase genes *alkB1*, *alkB2* (degradation of short- and middle-chain alkanes, C8–C16), *almA* (degradation of long-chain alkane, C16–C36), catechol 2, 3 dioxygenase (degradation of aromatic hydrocarbons), PAH-RHD(GP) (degradation aromatic hydrocarbons), and *nahAC* (naphthalene-degrading gene), are synthesized by microorganisms responsible for the breakdown of major crude oil constituents [9,15].

*Lysinibacillus fusiformis*, a Gram-positive rod-shaped bacterium belonging to the *phylum Firmicutes* and the family *Bacillaceae*, includes several strains noted for their diverse biotechnological potential and ability to produce relevant molecules and enzymes, especially esterases and peptidases, for industrial applications [16]. *Lysinibacillus fusiformis* is well known for its sporulation, which allows it to survive under extreme abiotic conditions [17]. Several studies have highlighted that *L. fusiformis* strains possess several beneficial traits that ensure their robustness under different environmental conditions [18,19]. The plant growth-promoting and antimicrobial characteristics of *L. fusiformis* strains were also evaluated in several studies, which reported some typical properties, such as the production of siderophore and auxin, solubilizing, and mineralizing phosphate, as well as the production of lipopeptide fengycin [20]. *Lysinibacillus fusiformis* expresses genes involved in the biodegradation of organic compounds [20] and exhibits tolerance to harsh environmental conditions [21]. These beneficial traits make *L. fusiformis* a promising microbial agent for biotechnological applications such as bioremediation and plant disease control, which could address pressing global challenges. This study evaluated the ability of the plant-associated bacterium *L. fusiformis* MGMM7 to bioremediate and biodegrade environmental pollutants, such as azo dyes, crude oil, and byproducts. Additionally, we sequenced the whole genome of MGMM7 and compared it to related genomes of *L. fusiformis* strains, specifically those isolated from sources containing xenobiotic contaminants.

## 2. Materials and Methods

The bacterial strain MGMM7 was isolated from the rhizosphere of the winter wheat (*Triticum aestivum* L.) variety Universiada (Tatarstan, Russia) during the tillering stage of plant growth. Rhizosphere soil samples were collected in sterile conical centrifuge tubes from agricultural land near Kazan (Tatarstan, Russia). The isolated bacterial strain was screened using Kings’ B medium (K_2_HPO_4_ 1.5 g, 0.75 g; MgSO_4_ × 7H_2_O 1.5 g, glycerol, 10 mL, pH 7.2 ± 0.2 at 25 °C). The purified culture was stored in a 20% glycerol stock at −80 ± 1 °C. The bacterial strain MGMM7 was identified as *L. fusiformis* based on 16S rRNA and was deposited in NCBI GenBank under accession number PQ001577. *Lysinibacillus fusiformis* MGMM7 is available from the Laboratory of Molecular Genetics and Microbiology Methods under accession number MGMM7.

### 2.1. Cell Suspension Preparation of L. fusiformis

The bacterial suspension was prepared from the culture of *L. fusiformis* growth in Landy medium (g/L: glucose, 20; glutamic acid, 5.0; yeast extract, 20, glucose; K_2_HPO_4_, 1.0; MgSO_4_, 0.5; KCl, 0.5; CuSO_4_, 0.0016; Fe_2_(SO_4_)_3_, 0.0012; and MnSO_4_, 0.0004; pH 7.0) at 30 ± 1 °C and 150 rpm for 24 h with a final cell concentration of not less than 10^9^ CFU/mL.

### 2.2. Biosurfactant Production by L. fusiformis MGMM7

The ability of MGMM7 to produce biosurfactants was determined according to Satpute [22]. For this purpose, 1 mL cell suspension of MGMM7 was inoculated into Landy medium and incubated at 30 ± 1 °C and 180 rpm for 72 h. After incubation, the culture was centrifuged for 10 min at 5000 rpm and 5 ± 1 °C. The obtained supernatant was filtered using a 0.45 μm membrane HN nylon filter (Millex, Merck Millipore, Darmstadt, Germany). To confirm the presence of biosurfactant in the cell-free MGMM7 supernatant, an oil spread test was performed by adding 500 μL petroleum oil to a Petri dish amended with distilled water. Subsequently, 50 μL of cell-free MGMM7 supernatant was gently applied at the center of the petroleum oil surface. The presence of biosurfactants was confirmed by the formation of clear zones on plate surfaces. All experiments were performed in triplicate, and 0.1% sodium dodecyl sulfate (SDS) and phosphate-buffered saline (PBS) (37 mM NaCl, 2.7 mM KCl, 10 mM Na_2_HPO_4_, and 1.8 mM KH_2_PO_4_, pH 7.0) solutions were used as positive and negative controls, respectively.

### 2.3. Crude Petroleum Oil Degradation Ability

The ability of *L. fusiformis* to degrade crude petroleum oil was examined in a flask glass containing 25 mL of mineral salt (MS) medium (g/L: 1.0 K_2_HPO_4_, 0.4 KH_2_PO_4_, 0.5 NaCl, 0.1 (NH4)_2_SO_4_, 0.2 NaNO_3_, and 0.025 MgSO_4_ × 7H_2_O) amended with 1% crude oil as the sole carbon source. Both crude oil and MSM were separately autoclaved at 121 °C for 20 min before use. The cell suspension was inoculated at a ratio of 1:100 in MS medium containing 1% crude petroleum oil and incubated at 30 °C and 180 rpm for 21 days. An uninoculated MSM medium containing 1% crude petroleum oil was used as a control sample. The degradation ability was considered positive based on the removal of oil layers visually observed compared to the negative control. Three independent tests were performed for statistical analysis. The amount of degraded petroleum oil in the medium was determined by the gravimetric method proposed by Panda et al. [23].

### 2.4. Crude Petroleum Oil Degradation Ability in Soil Experiment

#### Soil Experiment

The experimental design is presented in Table 1. Pots were incubated for up to 91 days at room temperature and watered once per week to maintain a water content of around 35%. The soil mixture and crude petroleum were autoclaved at 121 °C for 20 min before use.

The residual oil content in the soil was calculated by the gravimetric method according to Xu and Obbard [24] by measuring the soil fraction mass after crude oil extraction. Extraction was carried out three times to ensure the completeness of the crude oil extract from the soil. The oil degradation rate (ODR) was calculated using the following formula:$$ODR(\%)=\left(\frac{w_{1}-w_{2}}{w_{2}}\right)\times 100$$
where *W*_1_ and *W*_2_ as crude oil content in soil before and after degradation, respectively.

### 2.5. Plant Survival Ability in Remediated Soil

A laboratory pot experiment was conducted to assess the germination rate of plants on soil remediated with crude oil by *L. fusiformis* MGMM7. Garden cress (*Lepidium sativum* L.) was chosen as the test plant for large-scale biomonitoring due to its sensitivity [25]. Fifty (50) seeds were planted in each pot group (Table 1) in three replicates per group. Pots were incubated in a climate chamber under controlled conditions: temperature, 28 °C; light intensity, 100%; and humidity, 70% for up to 21 days. The germination rate was analyzed as the proportion of sprouted seeds (PSS), expressed as a percentage of the total number of seeds (TNS) taken for germination as described in the following equation:$$GP(\%)=\left(\frac{PSS}{TNS}\right)\times 100$$

### 2.6. Phenol Degradation Ability of L. fusiformis MGMM7

The ability of *L. fusiformis* MGMM7 to degrade phenol was assessed in basal medium (BM) (g/L: K_2_HPO_4_, 1,6; KH_2_PO_4_, 0.4; NH_4_NO_3_, 0.5; MgSO_4_·7H_2_O, 0.2; CaCl_2_, 0.025, FeCl_3_, 0.0025; 0.5% glucose) amended with phenol (2 mg/mL) (BMP). A cell suspension of MGMM7 was inoculated into a 96-well plate containing fresh BMP medium to obtain a final OD_595_ of 0.05. Bacterial growth was monitored by measuring OD_595_ every hour for 50 h at 30 °C using a spectrophotometer microplate reader (Allsheng, Hangzhou, China, AMR-100). A basal medium lacking phenol was used as a blank. Eight replicates were used for each condition for statistical analysis. Furthermore, the growth curve of MGMM7 in a BMP medium supplemented with phenol was compared with that of MGMM7 grown in BM without phenol under the same conditions.

Phenol degradation was measured according to Mahgoub et al. [26] by analyzing the UV-vis absorption spectra of culture broths at wavelengths between 196 and 380 nm. For this purpose, the basal medium obtained above was used.

### 2.7. Azo Dye Degradation Ability

The ability of *L. fusiformis* MGMM7 to degrade azo dye was assayed in MSM medium [K_2_HPO_4_, 1.73 g; KH_2_PO_4_, 0.68 g; MgSO_4_ 7H_2_O, 0.1 g; FeSO_4_-7H_2_O, 0.03 g; NH_4_NO_3_, 1 g; CaCl_2_·2H_2_O, 0.02 g; and 0.1% (*w*/*v*) of yeast extract] amended with 200 mg/L Congo red (CR) [1-naphthalenesulfonic acid, 3,3′-([1,1′-biphenyl]-4,4′-diyldiazo) bis-(4-amino-, disodium salt)]. A cell suspension of *L. fusiformis* MGMM7 at a ratio of 1:100 was inoculated into a 250 mL glass laboratory bottle containing 25 mL of fresh MSM medium then incubated at 30 ± 1 °C under the condition presented in Table 2.

All experiments were performed in triplicate. The dye-uninoculated and dye-free media were used as positive controls or blank and were also maintained under the same incubation conditions (Table 2). To quantify Congo red degradation, cultures were centrifuged at 10,000 rpm for 15 min, and the absorbance of the residual dye in the supernatant was assayed. The degradation percentage (DP) was measured by UV-Vis absorbance spectrometry using the following equation [27,28]:$$DP(\%)=\left(\frac{w_{1}-w_{2}}{w_{1}}\right)\times 100$$
where *W*_1_ and *W*_2_ as Congo red UV-Vis absorbance wavelengths before and after incubation, respectively.

### 2.8. Functional Genomic Analysis for Genetic Insights

#### 2.8.1. Genome Sequencing and Assembly

The genomic DNA of *L. fusiformis* MGMM7 was extracted using a phenol–chloroform extraction method [29] and cleaned using a cleanup kit (Evrogen, Moscow, Russia) according to the manufacturer’s instructions. The genomes were sequenced using an Illumina HiSeq 2500 System with 2 × 150 base pairs (bp) paired end reads. Raw reads with low sequencing quality and adapters were trimmed using Trimmomatic v. 0.36 and FastP v0.23.4-2 [30,31]. The genome was assembled using Spades Genome Assembler v3.15.4 [32]. JSpeciesWS (https://jspecies.ribohost.com/jspeciesws/#home, accessed on 20 March 2023) was based on a Tetra correlation search based on ANIb (average nucleotide identity based on BLAST+ v2.2.29). A contig scaffolding tool based on algebraic rearrangements (CSAR) [33] was used to scaffold contig to the reference genome. Plasmid Spades was used to identify the presence of plasmid contig in *L. fusiformis* MGMM7 [34]. Mob-recon v3.1.9 [35] was applied to identify plasmids in pre-assembled genomic contig generated by SPAdes v3.15.4. The NCBI Prokaryotic Genome Annotation Pipeline (PGAP) was used to annotate *L. fusiformis* strain MGMM7 [36].

#### 2.8.2. Genomic Comparison of *L. fusiformis* Strains

Genomic comparison compared it to related genomes of different *L. fusiformis* strains, specifically those isolated from relevant sources that degrade petroleum hydrocarbons. To assign each assembly to one of the two species, the average nucleotide identity (ANI) was calculated with the NCBI reference genomes using fastANI v1.32 (threshold higher or equal to 95.0% as the species identity). In silico DNA–DNA hybridization using the GGDC 3.0 server (https://ggdc.dsmz.de/home.php, accessed on 7 August 2024) was further used to validate their relatedness. The assigned assemblies’ genome species used in this study were downloaded from NCBI GenBank based on the following criteria: Completeness of the genome assemblies should not be below 90% and contamination should not exceed 10% using checkM2 v0.1.3 [37]. The selected genomes presented in Table 3 were used for downstream analyses.

#### 2.8.3. Comparison of the Genomic Characteristics of *L. fusiformis* Strains

Genome annotation and functional assignment were performed using Rapid Annotations using Subsystems Technology (RAST) to identify protein-coding sequences (CDS), rRNAs, tRNA genes, and subsystems. Annotated protein sequences were subsequently assigned to clusters of orthologous groups (COGs) using the RAST server and visualized as heatmaps generated by ClusVis v2.0 [39]. Furthermore, BlastKOALA v3.0 [40] was used for the functional genomic characterization of azo dye, crude oil, and byproduct degradation within the chromosomal and plasmid DNA of *L. fusiformis* strains. The secondary metabolites produced by *L. fusiformis* strains were analyzed using antiSMASH 7.0 with default parameters and relaxed detection strictness [41,42].

#### 2.8.4. Genomic Islands Analysis and Pan-Genome Reconstruction

Genomic islands in *L. fusiformis* were predicted using IslandViewer 4 [43]. Pangenome analysis was conducted to identify unique genes among the *L. fusiformis* strains. Genomes were initially annotated using Pokka v1.14.5 [44], and subsequent pan-genome analysis was performed using Roary v3.11.2 [45] on the generated GFF3 files. Gene clusters were categorized into core (present in 99–100% of strains), softcore (95–99%), shell (15–95%), and cloud (0–15%). A BLASTp identity threshold of 75% was applied. To exclude homologous and unique genes, synteny analysis was conducted using BLAST+ 2.9.0 [46], and RAST (Rapid Annotation using Subsystem Technology) server 2.0 [47]. Functional annotation of core and unique genes was performed using eggNOG-mapper v2.0 and blast2go v6.0.1 based on orthology predictions [48,49].

##### Statistical Analysis

Statistical analyses were performed using OriginLab Pro SR1 b9.5.1.195 (OriginLab Corp., Northampton, MA, USA). One-way ANOVA and Dunnett’s tests were used to estimate the significance levels at *p* < 0.05. The results are presented as the mean values ± standard deviation (SD).

## 3. Results

### 3.1. Screening for Biosurfactant Production by L. fusiformis MGMM7

The ability of *L. fusiformis* MGMM7 to produce a biosurfactant was assessed using oil spread tests, and the result is shown in Figure 1. As can be observed, the cell-free supernatant obtained from the culture of MGMM7 exhibited positive oil spread, indicating biosurfactant production (Figure 1B). After inoculation, clear zones with diameters ranging from 3.76 ± 0.086 to 4.927 ± 0.143 cm were immediately observed on Petri dishes. In contrast, the negative control (PBS solution) did not generate any clear zones (Figure 1A). Furthermore, the size of the clear zones created by the MGMM7 free-cell supernatant statistically did not differ from that of the positive control (SDS solution) (Figure 1C), further supporting the presence of MGMM7 free-cell supernatant of MGMM7.

### 3.2. Crude Oil Degradation Ability

The ability of MGMM7 to degrade crude oil components is shown in Figure 2. As can be observed, after 21 days of incubation, the oil coating area on the surface of the MS medium in Erlenmeyer flasks can be seen in the control setup (Figure 2A) in contrast to the inoculated experimental medium (Figure 2B). This indicates that *L. fusiformis* MGMM7 can degrade crude petroleum. Quantitative determination of crude oil degradation using the gravimetric method revealed that *L. fusiformis* MGMM7 can degrade crude oil by up to 44.55 ± 5.47% in MS medium.

### 3.3. Crude Oil Degradation in Contaminated Soil Experiment

The degradation ability of MGMM7 in the contaminated soil experiment was evaluated after 3 months of incubation at room temperature. The results indicate that the MGMM7 strain can degrade crude oil. The degradation rate was assayed as 51.31 ± 2.57%.

### 3.4. Survival Rates of Plants in Remediated Soil

The results obtained are shown in Figure 3. After incubation, it was observed that treating crude oil soil contaminated with *L. fusiformis* MGMM7 increased seed germination. The seed germination in soil treated with the strain increased by up to 43.61 ± 0.67% compared with plans for group 2 [garden soil amended with crude oil at the ratio of 1:1 (m/m)]. The germination rate of plants grown in soil amended with crude oil was measured as 20.83 ± 2.06%. In the control group (group 1, oil-free soil), seed germination was 86.55 ± 3.51%. Moreover, adequate coverage of plant roots in group 3 pretreated with MGMM7 strain as compared with group 2 was observed (Figure 3).

### 3.5. Lysinibacillus fusiformis MGMM7 Growth in Phenol and Its Degradation Ability

The MGMM7 growth curves for phenol degradation are shown in Figure 4. As can be seen, the growth strain in both cultures immediately displayed a logarithmic phase within 1 h after inoculation, with no statistically significant difference (*p* < 0.05) compared with the control samples. This indicates the phenol tolerance and degradation ability of MGMM7. The logarithmic phase lasted approximately 34 h (Figure 4A). The strain MGMM7 in BM registered a stationary phase due to nutrient depletion, whereas the culture with phenol (BMP) continued to grow. The analysis of the absorption spectra at 208 and 270 nm revealed a decrease in the phenol concentration after 48 h of incubation (Figure 4B). The absorption spectra of the free culture of MGMM7 after incubation showed a decrease in the concentration of phenol at wavelengths 208 and 270 nm. The UV-vis spectra of phenol at initial concentrations (2 mg/mL) were evaluated as 29.68 ± 0.91 and 24.25 ± 1.05, while after the incubation, as 24.31 ± 2.43 and 15.11 ± 3.07 at 208 and 270 nm, respectively.

### 3.6. Azo Dye Degradation Ability of L. fusiformis MGMM7

The ability of *L. fusiformis* MGMM7 to discolorize and degrade Congo Red (CR) azo dye was evaluated under laboratory-controlled conditions. After incubation for 24 h, discoloration of the medium in both shaking and static conditions was observed compared with the control (Figure 5A). The UV-vis absorption spectra analysis showed a statistically significant difference in the decrease of CR azo dye absorbance (Figure 5B). The absorption spectrum under static conditions exhibited a 1.6-fold ability to degrade CR azo dye as compared to *L. fusiformis* MGMM7 incubated at 150 rpm, which decreased from 5.12 ± 0.13 (control) to 3.10 ± 0.18 and 1.92 ± 0.10 absorbance units under shaking and static culture conditions, respectively. The ability of *L. fusiformis* MGMM7 to degrade 200 mg/L of CR azo dye was assayed as 39.80 ± 0.17% and 62.32 ± 0.16% under shaking and static culture conditions, respectively, at 497 nm.

A similar result was observed after 72 h of incubation (Figure 6). Biodegradation under static culture conditions of strain *L. fusiformis* MGMM7 statistically was up to 4.5 ± 0.7% effective in transforming CR azo dye, as compared under shaking and static + shaking culture conditions. Under static cultivation conditions, a significant CR azo dye discoloration was observed (Figure 6A). The absorption spectrum of CR azo dye medium static conditions of *L. fusiformis* MGMM7 was assayed as 1.15, whereas these under shaking, static + shaking, and control groups were assayed as 1.53, 1.38, and 5.12 abs, respectively (Figure 6B). The ability of *L. fusiformis* MGMM7 to transform Congo Red (CR) azo dye (200 mg/mL) under tested conditions was evaluated as 77.53 ± 1.7%, 73.04 ± 0.8%, and 70.71 ± 1.0%, static, static + shaking, shaking culture conditions, respectively, for 72 h at 30 ± 1 °C. Additionally, under tested conditions, the pH levels before and after cultivation ranged from 7.0 to 8.0 with a variance of ±1 °C.

### 3.7. Genome Assembly of L. fusiformis MGMM7

The assembly of *L. fusiformis* MGMM7 generated a chromosome and circular plasmid that comprised 5,028,939 and 244,822 bp, respectively, with G + C contents of 37.34% and 36.92%. A Tetra correlation search revealed that MGMM7 was closely related to *L. fusiformis* M5 (NCBI RefSeq assembly: GCF_001726065.1; BioSample ID: SAMN05715906), with 98.03% similarity based on ANIb (average nucleotide identity based on BLAST). Automated annotation using PGAP revealed that MGMM7 carries a total of 5267 genes, including 5043 coding genes. *Lysinibacillus fusiformis* MGMM7 contains 162 genes (RNA), 119 of which are transfer ribonucleic acid (tRNA) and 6 are non-coding RNA (ncRNA). The genome assembly was deposited in NCBI GenBank under accession GenBank: CP130331.1.

#### 3.7.1. Genome Comparison of *L. fusiformis* Strains

Comparative genomic analysis revealed that MGMM7, a strain isolated from winter wheat rhizospheres, exhibited an average nucleotide identity (ANI) of less than 96% with strains Cu1-5, HJ.T1, and TZA38 isolated from sludge, polyester fabric, and sinkhole, respectively (Figure 7). The ANI values of *L. fusiformis* strains TZA38, Cu1-5, and HJ.T1 were high, exceeding 98% (Figure 7A). Similar results were obtained from in silico digital DNA–DNA hybridization (dDDH) analyses based on generalized linear models (GLM). dDDH estimations revealed a distinct pattern. *L. fusiformis* MGMM7 shared an identity value ranging from 64.30% to 64.80% with strains Cu1-5, HJ.T1, and TZA38. *The L. fusiformis* strains Cu1-5, HJ.T1, and TZA38 exhibited dDDH values of up to 92.30% (Figure 7B).

#### 3.7.2. Comparison of the Genomic Characteristics of *L. fusiformis* Strains

The genomic characteristics of the *L*. *fusiformis* strains used in this study are presented in Table 4. The analyzed *L. fusiformis* strains exhibited chromosome lengths ranging from 4.51 to 5.02 Mbp and GC contents between 37.34% and 37.5%. The analyzed strains carried a coding sequence (CDs) ranging from 4406 to 5105. The percentage of pseudogenes ranged from 1.03% to 1.95%, with *L. fusiformis* Cu1-5 isolated from sludge exhibiting the highest pseudogene content. Variations in genome size, gene content, and pseudogene frequency indicate substantial differences in gene gains and losses during the evolution of these strains.

#### 3.7.3. Specification of Secondary Metabolite Biosynthesis Gene Clusters in *L. fusiformis* Strains

The in silico analysis of the secondary metabolite in *L. fusiformis* using antiSMASH 7.0 revealed the presence of six gene clusters (Appendix A), of which four were predicted to be involved in the biosynthesis of metabolites, NRPS-like (non-ribosomal peptide synthetase-like fragment), T3PKS (polyketide synthase type III), betalactone, and NI-siderophore (NRPS-independent, IucA/IucC-like siderophores), which encode kijanimicin, bacillibactin/bacillibactin E/bacillibactin F, fengycin, and aerobactin, respectively (Appendix A). Two biosynthetic gene clusters that did not match the most known metabolites in the databases were predicted for MGMM7, Cu1-5, HJ.T1, and TZA38. Additionally, a biosynthetic gene cluster associated with the synthesis of a non-ribosomal peptide molybdenum cofactor predicted in *L. fusiformis* MGMM7 was absent in Cu1-5, HJ.T1, and TZA38.

#### 3.7.4. Specifications of Genes Involved in Xenobiotic Degradation in Plasmids

Plasmid presence was predicted in HJ.T1 and MGMM7. The analyzed plasmids harbored varying percentages of pseudogenes, which varied from 0.80% to 1.01%, with the highest rate observed in the HJ.T1 plasmid isolated from polyester fabric (Table 4). Plasmids predicted in MGMM7 carried genes related to the synthesis of proteins involved in the degradation of azo dye and crude oil. These include *mhqO* (putative ring-cleaving dioxygenase), *hmoA* (heme-degrading monooxygenase), o-succinylbenzoate synthase, *azoR4* (FMN-dependent NADH-azoreductase 4), *cat1* (succinyl-CoA: coenzyme A transferase), *yclQ* (petrobactin-binding protein), and *sphR* (alkaline phosphatase). In addition, the presence of genes responsible for metal transport, such as iron-uptake system-binding proteins, zinc transporters, and other metals, was identified in MGMM7 (Figure 7). In contrast, the plasmid of *L. fusiformis* HJ.T1, isolated from polyester fabric, lacked genes associated with oil degradation or azo dye metabolism. The complete nucleotide sequence of the MGMM7 plasmid was deposited in NCBI GenBank under accession number NZ_CP130332.1.

#### 3.7.5. Subsystem Comparison of Functional Gene Groups

Comparative genomics analysis using RAST server revealed substantial differences in genes associated with subsystem categories among the analyzed strains (Figure 8). As depicted in the dendrogram, *L. fusiformis* MGMM7, isolated from wheat rhizospheres, exhibited a higher number of subsystems than Cu1-5, HJ.T1, and TZA38 strains originating from sludge, polyester fabric, and sinkholes, respectively (Figure 8). *Lysinibacillus fusiformis* MGMM7 carried more genes connected to subsystems involved in fatty acid and isoprenoid metabolism, nucleotide metabolism, amino acid metabolism, cofactor and vitamin biosynthesis, protein metabolism, dormancy and sporulation, RNA metabolism, virulence, disease, and defense. In contrast, L. *fusiformis* Cu1-5, HJ.T1, and TZA38 carried more genes related to subsystems involved in carbohydrate and phosphorus metabolism than MGMM7. No significant differences were observed in genes associated with subsystems involved in aromatic compounds, nitrogen metabolism, and iron acquisition pathways.

#### 3.7.6. Pan-Genome Reconstruction

Pangenome analysis using Roary identified 3884 core and 2040 shell genes among the *L. fusiformis* strains, representing a total of 5924 genes (Figure 9A). *L. fusiformis* MGMM7 exhibited a large repertoire of unique genes (cspLA, nemA, bpoC, farB, craA, phnT, kamA, srfAD, tycC, pksJ, sfp, uviB, hisE, hpaIIM, rtp, tylCV, glcU, speH, ydjZ, bluB, dapX, mdaB, yceM, hrtA, ymfD, yjaB, ydeA, hcaR, phnC, phnE, ywqF, gmd, fcl, rfbM, kdsB, pseI, pseC, pgcA, rfbE, fdtA, fdtC, fdtB, rmlA, rmlC, rfbB, rmlD, wbbL, and wcaJ) compared to TZA38, Cu1-5, and HJ.T1 (Figure 9B). The predicted unique genes were categorized into COG functional groups S (unknown function), E (amino acid transport and metabolism), M (cell wall/membrane/envelope biogenesis), T (signal transduction mechanisms), C (energy production and conversion), V (defense mechanisms), P (inorganic ion transport and metabolism), U (intracellular trafficking, secretion, and vesicular transport), J (translation, ribosomal structure and biogenesis), and G (carbohydrate transport and metabolism) based on eggNOG mapper annotation. For example, among the predicted genes, nemA participates in the degradation of toxic nitrous compounds besides N-ethylmaleimide, bluB is involved in the biosynthesis of cobalamin, and phnCET encodes proteins necessary for the use of phosphonate as the sole phosphorus source. Notably, *L. fusiformis* strain HJ.T1 possessed 11 genes (*mhpE*, *cysQ*, *ant1*, *essG*, *yeeF*, *cdiA*, *dpnM*, *ltxD*, *lagD*, *cwlK*, and *wapA*), which were absent in the other strains (Figure 9C). *L. fusiformis* TZA38 and Cu1-5 carried eight (*betB*, *pglF*, *epsD*, *thiK*, *patA*, *recT*, *ramB*, *cpdR*) and three (*ywqK*, *lutP*, *neuB*) unique genes, respectively.

### 3.8. Comparison of Genes Involved in Xenobiotic DNA Degradation

Genomic comparative analysis based on automatic KO assignment and the KEGG mapping service demonstrated a conserved gene profile related to xenobiotic degradation among the *L. fusiformis* strains (Appendix A). The analyzed genomes of TZA38, HJ.T1, MGMM7, and Cu1-5 encoded 4-oxalocrotonate tautomerase, which is involved in the degradation of toluene, o-xylene, 3-ethyltoluene, and 1,2,4-trimethylbenzene. Genes associated with the synthesis of biphenyl-2,3-diol-1,2-dioxygenase, which plays a key role in the degradation of biphenyl and gamma-hexachlorocyclohexane, were also predicted in the genomes of TZA38, HJ.T1, MGMM7, and Cu1-5. A variety of genes encoding monooxygenase, dioxygenase, and dehydrogenases, such as 4-hydroxyphenylacetate 3-monooxygenase, catechol 2,3-dioxygenase, ring-cleaving dioxygenase, and cyclopentanol dehydrogenase, were identified in MGMM7 and the other strains. Genes associated with naphthalene degradation, such as alcohol dehydrogenase and S-(hydroxymethyl)glutathione dehydrogenase, were also conserved across all strains. Furthermore, all genomes encode genes involved in the synthesis of azoreductase. The presence of genes involved in the degradation of aminobenzoates, such as amidase, was specifically identified in the *L. fusiformis* strains MGMM7 and Cu1-5. In contrast, no genes associated with the synthesis of alkane monooxygenase (*AlkB*), rubredoxin reductase (*AlkT*), and rubredoxin-2 (*AlkG*), which are involved in the first stage of the degradation process of n-alkanes, were predicted in the studied *L. fusiformis* strains. However, genes involved in the second stage, such as aldehyde dehydrogenase (*alkH*) and alcohol dehydrogenase (*alkJ*), were present in the genomes of TZA38, HJ.T1, MGMM7, and Cu1-5.

### 3.9. Specifications of Genes Involved in Xenobiotic Degradation in Genomic Islands (GIs)

Genomic islands are large sections predicted in a genome harboring genes involved in a symbiosis that usually exhibit signatures of gene clusters acquired through horizontal gene transfer. In this section, genes were analyzed and compared between *L. fusiformis* strains. The obtained results identified 10, 12, 23, and 8 genomic islands (GIs) in the genomes of TZA38, HJ.T1, MGMM7, and Cu1-5, respectively (Figure 9, Appendix A). These gene clusters comprised 2.56%, 4.56%, 8.89%, and 4.56% of the corresponding genomes. The annotations of this gene cluster are detailed in Appendix A.

The genomic islands (GIs) of *L. fusiformis* strains TZA38, HJ.T1, MGMM7, and Cu1-5 carried a high proportion of genes with unknown functions (79.24%, 78.27%, 73.93%, and 75.62%, respectively). The remaining percentage of genes in these strains encode proteins involved in diverse cellular processes. For example, a section predicted in Cu1-5 between 1,220,355 and 1,228,051 bp (Figure 10, Appendix A) contained genes related to alkaline phosphatase synthesis (*phoP*, *phoR*) and energy metabolism (dephospho-CoA kinase, glyceraldehyde-3-phosphate dehydrogenase2). Another section predicted in Cu1-5 (1,953,056–2,011,987 bp) harbored genes associated with the oxidative stress response (*ohrA*), metal efflux (*mneS*), and redox reactions (NAD(P)H nitroreductase, *ftsH*). Additionally, a section predicted in the region 2,590,530–2,601,758 bp of Cu1-5-encoded proteins involved in haloalkane degradation and lipid metabolism (haloalkane dehalogenase, 4-dihydroxy-2-naphthoate octaprenyltransferase, zinc-binding alcohol dehydrogenase).

The genome of TZA38 harboring several genes with unknown functions showed fewer genes involved in the degradation of xenobiotic compounds within its genomic islands (GIs). Nevertheless, the presence of enzyme-encoding genes involved in deoxyribose-phosphate biosynthesis (deoxyribose-phosphate aldolase), metal transport (*corA*), chemotaxis (*hemAT*), and one-carbon metabolism (carboxy-S-adenosyl-L-methionine synthase) in its GI between positions 467,490 and 485,283 bp. Additionally, TZA38 harbored large sections containing genes related to lipid metabolism (3-hydroxy-3-methylglutaryl coenzyme A reductase), calcium transport (calcium-transporting ATPase 1), and mobile genetic elements (IS1595 family transposase ISSpgl1) within regions 2,667,904–2,675,522 bp, 3,150,655–3,157,941 bp, and 4,350,820–4,372,441 bp, respectively.

Similarly, HJ.T1 carried a large section between positions 605,213 and 635,687 bp that encoded genes involved in pyridoxal metabolism (pyridoxal 4-dehydrogenase) and mobile genetic elements (IS3 family transposases ISBth167, ISBce15, and ISBko1). Genes involved in energy metabolism (succinate dehydrogenase cytochrome b558 subunit, fumarate reductase iron–sulfur subunit) were found within the group with GI between 3,395,848 and 3,418,441 bp of HJ.T1.

The presence of multiple large sections acquired through horizontal gene transfer that encodes genes with diverse metabolic functions was predicted in the MGMM7 genome. A section spanning the genome between 2,215,552 and 2,255,220 bp contains genes involved in aromatic compound biosynthesis (2-succinyl-6-hydroxy-2,4-cyclohexadiene-1-carboxylate synthase), metal homeostasis (multicopper oxidase (*MCO*), arsenical resistance protein (*acr3)*, arsenate reductase, zinc transporter (*zupT*), manganese efflux system protein (*mneS*), redox reactions (ferredoxin-NADP reductase, NADPH: quinone oxidoreductase *mdab*), and nucleotide metabolism (purine efflux pump *pbuE*, general stress protein 17M) (Figure 10, Appendix A). A large section between 2,424,983 and 2,431,174 bp predicted in the genome of MGMM7 carried genes related to the synthesis of 7-carboxy-7-deazaguanine synthase, 7-cyano-7-deazaguanine synthase, and NADPH-dependent 7-cyano-7-deazaguanine reductase. The presence of the transposable element IS1595 transposase (*ISCac2*) was also observed in this region. Another section between 4,888,110 and 5,021,990 bp was predicted to harbor genes such as catechol-2,3-dioxygenase, which are involved in the degradation of aromatic compounds; L-aspartate oxidase, glutamate dehydrogenase genes, which are involved in the metabolism of amino acids; long-chain alcohol dehydrogenase 2 genes, which are involved in lipid metabolism; as well as genes encoding the synthesis of nicotinate-nucleotide pyrophosphorylase.

## 4. Discussion

Bioremediation is performed by microbes, which are usually isolated from contaminated sites. Active microbial metabolite production and genetic systems of xenobiotic degradation give these microbes advantages to survive and even proliferation in polluted ecological niches. In this study, we characterized *L. fusiformis* MGMM7, a strain isolated not from a contaminated environment but from the rhizosphere of winter wheat (*Triticum aestivum* L.), to determine its ability to degrade crude oil, phenol as a byproduct, and azo dyes, and compared it with other strains. There have been several reports of *L. fusiformis* strains to produce biosurfactant and degrade petroleum waste, whereby the strains were isolated from automobile-mechanic-workshop soil, mechanical village, and wastewater and were able to degrade PAHs [50,51,52,53]. To evaluate the ability of MGMM7 to degrade crude oil and its products, we initially tested the efficacy of biosurfactants using a crude oil dispersion test. The test showed oil dispersion of MGMM7 3.76 ± 0.086 to 4.927 ± 0.143 cm (Figure 1) in the control sample with 0.1% SDS. This shows the ability of MGMM7 to not only produce biosurfactants but also transform crude oil into droplets based on their high surface activity, thereby increasing the surface area for microbes to utilize the oil, especially in contaminated waters [54]. It has been reported that surfactants produced by microorganisms improve hydrocarbon assimilation [55,56]. By increasing the solubility of PAHs, biosurfactants facilitate their accessibility to PAH-degrading microorganisms [57]. Two types of low-molecular-weight biosurfactants (glycolipids, lipopeptides, and phospholipids) are responsible for reducing interfacial surface tension and enhancing the solubilization of hydrophobic PAHs [58]. Based on the above information, we annotated the gene clusters responsible for producing the low-molecular-weight lipopeptide biosurfactant and polyketide antibiotic MGMM7. These lipopeptide biosurfactants can have antibiotic activities that improve the robustness of microbes through their antibiosis mechanisms. Although the percentage identity to their known gene clusters is low, the kijanimicin gene cluster may synthesize new antibiotic compounds, as stated by Zhou et al. [57]. Another interesting result was the presence of Petrobactin, which was also stated in the above manuscript as an oxidative stress alleviation compound and enhances spore formation in *Bacillus anthracis* [58]. In our study, we were able to predict the gene responsible for its transmembrane transfer, known as *yclQ* (petrobactin-binding protein). This attests to not only the expression of genes responsible for the synthesis of petrobactin but also the presence of the ABC transporter complex protein responsible for its transport.

The ability of *L. fusiformis* MGMM7 to degrade components of crude oil evaluated through gravimetric analysis showed a percentage of 44.55 ± 5.47% (Figure 2) in 21 days, whereas *Lysinibacillus* sp. SS1 [59] showed 84.30 ± 0.13% degradation of components of crude oil in 28 days. This may be due to the source from which *Lysinibacillus* sp. SS1 was isolated from the oil-contaminated soil of an automotive station, which might contribute to its adaptability and robustness in this environment and contribute to its high degradation rate. Although MGMM7 was isolated from wheat rhizospheres and lacks genes associated with the initial stages of aerobic alkane assimilation [alkane monooxygenase (*alkB*), rubredoxin reductase (*alkT)*, and rubredoxin-2 (*alkG*)], it exhibited a notable ability to degrade some components of crude oil [60,61]. This may be due to the presence of luciferase-like monooxygenase (LLM) flavin-dependent oxidoreductase, a member of the flavin-containing monooxygenase superfamily involved in alkane catabolism, which may contribute to the ability of *L. fusiformis* MGMM7 to cleave alkanes. As reported by Shi-Weng [62], a gene encoding LLM flavin-dependent oxidoreductase shares a similar functionality with alkane 1-monooxygenase. Likewise, strain 15-4 isolated from uncontaminated soil in a cold region on the Qinghai–Tibet Plateau demonstrated a rapid hydrocarbon degradation rate of 56.64% at 20 °C within 96 h of incubation [62]. Transcriptome analysis after inoculation in petroleum did show the expression of genes responsible for alkane transfer, hydroxylation, and aromatic compound degradation, which are also present in MGMM7, although both strains have not been isolated from contaminated sources. A similar result was reported by Nithimethachoke et al. [61] who demonstrated that thermophilic *Geobacillus kaustophilus* HTA426 lacking an alkane oxygenating enzyme could degrade alkanes. The ability was associated with the presence of the putative ribonucleotide reductase small subunit GkR2loxI, which encodes a novel alkane monooxygenase/hydroxylase with heterodinuclear Mn–Fe.

The degradation of hydrocarbons produces fatty acids with different lengths of carbon chains, which are then activated by fatty acid-CoA synthetases to form fatty acyl-CoA derivatives [63,64]. The presence of genes involved in fatty acid metabolism, such as fatty acid-CoA synthase, long-chain fatty acid-CoA synthetase, acetyl-CoA synthetase, and acetyl-CoA acetyltransferase, was predicted in all the analyzed *L. fusiformis* strains, suggesting their ability to activate fatty acid genes by converting them to fatty acyl-CoA derivatives for subsequent β-oxidation. Additionally, the strain MGMM7’s ability to degrade crude oil components in contaminated soil also reduced crude oil toxicity to plants by the experimental setup after 3 months of inoculation. As stated above, garden cress (*Lepidium sativum* L.) is sensitive to both abiotic and biotic stress and considers a germination rate of 65% to control samples without inoculation (20.83%). Rhizoremediation, which is based on the interaction between plant roots and rhizobacterium, has been investigated as a low-cost and effective technology for the complete mineralization of pollutants from the environment [65]. The ability of MGMM7 to process crude oil components and promote plant growth indicates its potential for biotechnological applications in the rhizoremediation of petroleum–organic pollutants. Most microorganisms with high phenol-degrading ability have been isolated from polluted environments. Bacteria *Halomonas* sp. strain PH2-2, *Acinetobacter* sp. EDP3, and *Alcaligenes faecalis* isolated from hypersaline wastewaters, gray water bio-processor, and industrial-activated sludge, respectively, were reported to degrade up to 1100 mg/L phenol and use phenol as its sole carbon source [66,67,68]. In the initial degradation, phenol undergoes monohydroxylation at the ortho position, adjacent to its existing hydroxyl group, facilitated by the enzyme phenol hydroxylase resulting in the formation of catechol. Subsequently, catechol can be acted upon by one of two enzymes: catechol 1,2-dioxygenase, which triggers the ortho pathway, leading to the formation of succinic acid and acetyl-CoA [69]; or catechol 2,3-dioxygenase, which initiates the meta pathway, resulting in the production of pyruvate and acetaldehyde [70,71]. The experimental setup for phenol degradation proved the ability of bacterial strain MGMM7 isolated from the rhizosphere to use phenol as a carbon source for its growth (Figure 4). For the first 34 h, MGMM7 was able to utilize glucose first, and when there was the absence of any source of carbon, it entered the stationary state in contrast to the experimental sample switching to phenol as a source of carbon, noticing an exponential growth up to 50 h. This may be due to the presence of genes such as FAD-containing monooxygenase, 4-hydroxyphenylacetate 3-monooxygenase oxygenase allowing *L. fusiformis* MGMM7 to oxidize phenol to catechol via the β-ketoadipate pathway.

The biotransformation of azo dye generally requires a combination of two stages (Feng) [72]. The first stage usually occurs under both anaerobic and aerobic conditions and is responsible for breaking azo-band (-N=N-) between two or more aromatic rings, leading to the discoloration of dye and the formation of aromatic amines [73,74,75]. The second stage involved in the mineralization of amine groups [76] mostly occurs under aerobic culture conditions, since aromatic amines are dead-end metabolites of the anaerobic azo dyes biotransformation [77]. Several studies have shown that static conditions are preferred and are more efficient for azo dye discoloration [78,79]. In our study, *L. fusiformis* MGMM7 demonstrated a significant ability to degrade the CR azo dye (200 mg/L), achieving degradation rates of up to 60% under both static and shaking cultivation conditions. However, the highest degradation efficiency was observed under shaking conditions. Furthermore, the degradation of CR azo dyes by *L. fusiformis* MGMM7 was less effective under integrated (anaerobic and aerobic) culture conditions. A similar finding was previously reported by Sari and Simarani [80], in which *L. fusiformis* strain W1B6 exhibited a non-significant difference in the decolorization of azo dye methyl red under static and shaking conditions, achieving decolorization rates of up to 96% and 93.6%, respectively. In a study conducted by Maurya et al. [81], *L. fusiformis* KLM 1 isolated from a dye-contaminated site showed efficacy in reducing the CR content to 88.4% and 64.6% at concentrations of 50–250 mg/L for 113 days, respectively. Additionally, genomes of *L. fusiformis* MGMM7, along with those of TZA38, HJ.T1, and Cu1-5, carried azoreductase, 3-hydroxybutyryl-CoA dehydrogenase, salicylate hydroxylase, salicylaldehyde dehydrogenase, N-acetyltransferase, NADH quinone oxidoreductase, and NADH flavin reductase genes associated with in assimilation of azo dye as reported in the transcriptome analysis of degradative pathways for Azo Dye Acid Blue 113 in *Sphingmonas melonis* B-2 [82].

The average nucleotide identity has been revealed to correlate with digital DNA hybridization, with a range of approximately 95–96% ANI often corresponding to the current threshold of 70% DDH similarity [83]. According to Goris et al. [84], ANI values between 70% and 80% are considered sufficient for distinguishing strain relatedness. The results demonstrated that MGMM7 can reliably be distinguished from Cu1-5, HJT.1, and TZA38 isolated from sludge, polyester fabric, and sinkholes, respectively, by their digital DDH values, which were much lower than the 70% threshold. In contrast, the DDH values between Cu1-5, HJT.1, and TZA38 were calculated to be at least 89% (Figure 7B). These findings suggest a closer taxonomic relationship among the source-contaminated *L. fusiformis* strains analyzed in this study compared with MGMM7. Moreover, *L. fusiformis* MGMM7 carried more unique genes, as well as many subsystems involved in fatty acid and isoprenoid metabolism, nucleotide metabolism, amino acid metabolism, cofactor and vitamin biosynthesis, protein metabolism, virulence, and defense. These variations may be attributable to diverse nutritional factors, environmental adaptations, and ecological niche specialization [85,86]. For example, rhizosphere microorganisms are more exposed to various organic compounds from plant root secretions, which may contribute to the acquisition of accessory genes that facilitate niche-specific adaptation [87]. Furthermore, the dynamic and complex microenvironment of the rhizosphere can drive evolutionary changes in gene repertoires [88], whereas strains isolated from contaminated sources are more clustered for a limited range of substrates and conditions due to local abiotic conditions and biotic interactions [89,90].

Genomic islands are large sections or discrete cluster genes predicted in a genome-harboring gene involved in a symbiosis that usually exhibits signatures of gene clusters acquired through horizontal gene transfer [91,92]. Among the compared genomes, MGMM7 carried more genomic islands (GIs) with a more functional gene compared to Cu1-5, HJ.T1, and TZA38. This trend was also observed in the MGMM7 plasmid, which contained more genes associated with azo dye and PAH assimilation. This may be attributed to the fact that rhizosphere bacteria frequently engage in horizontal gene transfer, which leads to the acquisition of diverse genes [93,94], whereas strains isolated from contaminated sources may experience gene loss associated with non-essential functions [95,96].

## 5. Conclusions

The selection of microbes for hydrocarbon bioremediation often focuses on strains isolated from relevant contaminated sources. In this study, we demonstrated that *L. fusiformis* MGMM7, a representative microbial species with well-known hydrocarbon bioremediating ability isolated from an uncontaminated source wheat rhizosphere, can retain its crude oil-degrading assimilation. *Lysinibacillus fusiformis* MGMM7 has demonstrated the ability to remediate soil contaminated with crude oil and phenol and to degrade azo dye Congo red. Subsystem distribution using RAST SEED and pangenome analysis emphasized the distinctive features of MGMM7, including functional genes in its chromosome and plasmid as well as the presence of several unique genes associated with large functional GO term categories, such as those involved in PAH assimilation. Comprehensive genomic analysis and comparison with related strains isolated from contaminated sources revealed an almost identical genomic architecture. The rhizosphere strain MGMM7 carries more genetic islands within a unique set of functional genes. These findings suggest that incorporating rhizosphere strains into hydrocarbon bioremediation strategies can significantly benefit the rhizoremediation of petroleum hydrocarbons. Future research should delve deeper into the molecular mechanisms underlying the degradation of crude oil, phenol, and azo dyes by MGMM7. Additionally, large-scale field trials are necessary to evaluate the practical application of this strain in bioremediation and to assess its potential interactions with plants in the removal of pollutants from petroleum hydrocarbon-contaminated sites.

## Figures and Tables

**Figure 1 microorganisms-12-02377-f001:**
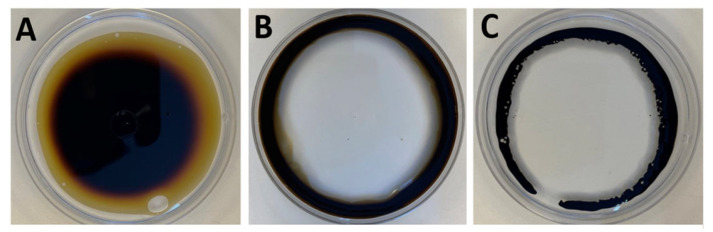
Biosurfactant production by *L. fusiformis* MGMM7. Inoculation of 50 µL of distilled water (**A**), free-cell supernatant of MGMM7 (**B**), and 0.1% SDS solution (**C**) in the surface petroleum oil into Petri dishes amended with distilled water.

**Figure 2 microorganisms-12-02377-f002:**
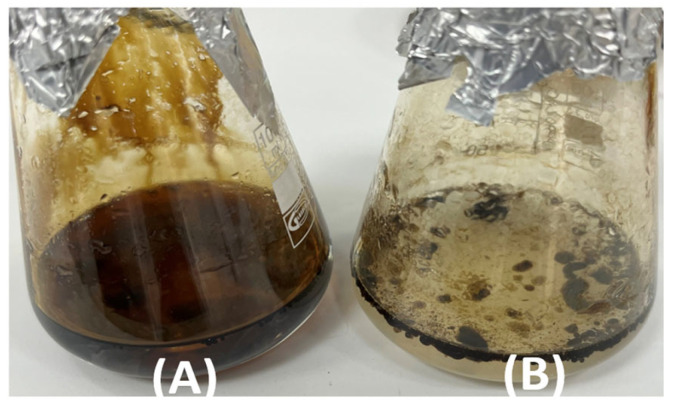
Qualitative analysis of crude oil degradation by *L. fusiformis* MGMM7. Control: MS medium amended with 1% crude petroleum (**A**); MS medium amended with 1% crude inoculated with *L. fusiformis* MGMM7 (**B**) after 21 days of incubation at 30 ± 1 °C.

**Figure 3 microorganisms-12-02377-f003:**
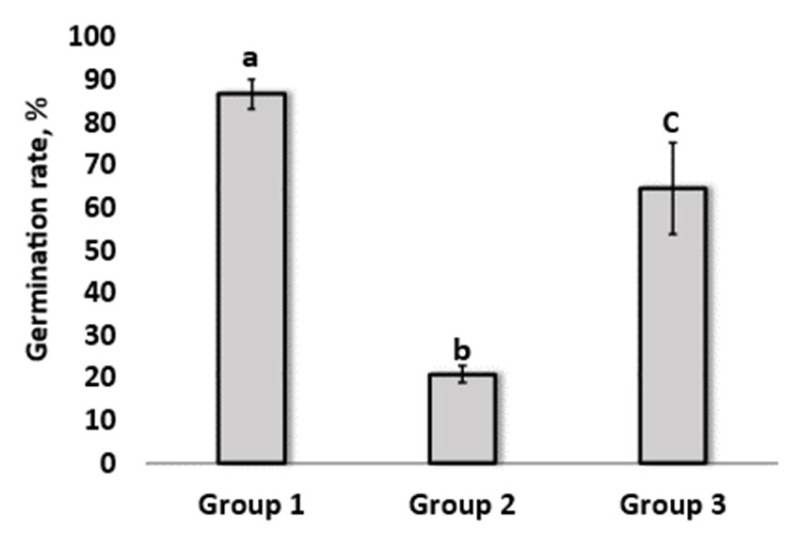
Remediation of oil-contaminated soil by *L. fusiformis* MGMM7 in pot trial experiment. Plant growth ability under different soil conditions. Garden cress (*Lepidium sativum* L.) plants were grown in a climate chamber under controlled conditions: temperature, 28 °C; light intensity, 100%; and humidity, 70% for up to 21 days. The results are expressed as mean values ± standard deviation (SD). One-way ANOVA and Dunnett’s tests were used to estimate the significance levels at *p* < 0.05. Statistical difference among groups is labeled with different letters.

**Figure 4 microorganisms-12-02377-f004:**
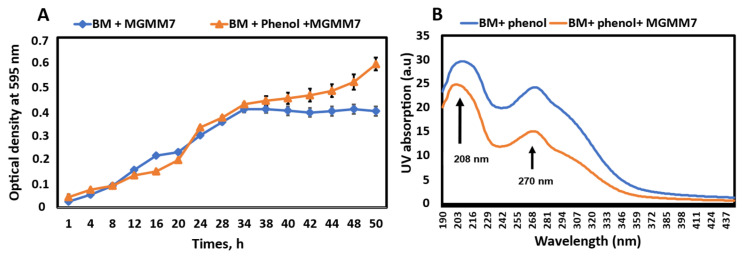
Growth and degradation ability of *L. fusiformis* MGMM7 in basal medium amended with or without phenol (2 mg/mL) at 30 ± 1 °C for 50 h (**A**) and UV-vis absorption spectra after 50 h of the incubation period (**B**).

**Figure 5 microorganisms-12-02377-f005:**
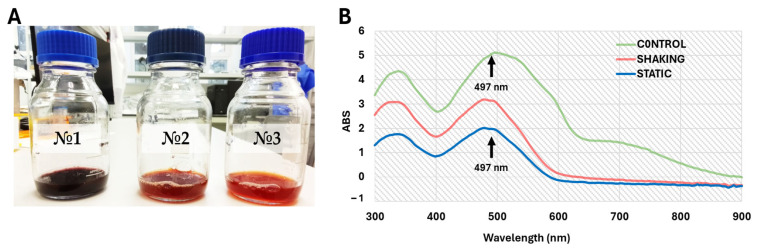
Biotransformation of CR azo dye by *L. fusiformis* MGMM7 at 30 ± 1 °C for 24 h. Discoloration of CR azo dye (**A**) and UV-vis absorption spectra (**B**). Control sample (No. 1); CR azo dye degradation by *L. fusiformis* MGMM7 under shaking (No. 2) and static (No. 3) culture conditions.

**Figure 6 microorganisms-12-02377-f006:**
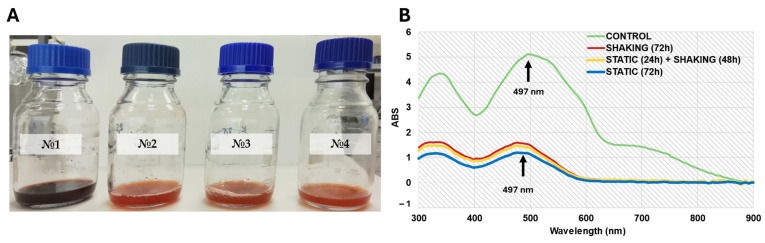
Biotransformation of CR azo dye by *L. fusiformis* MGMM7 at 30 ± 1 °C for 72 h. Discoloration of CR azo dye (**A**) and UV-vis absorption spectra (**B**). Control sample (No. 1); CR azo dye biotransformation by *L. fusiformis* MGMM7 under shaking (No. 2), static + shaking (No. 3), and static (No. 4) culture conditions.

**Figure 7 microorganisms-12-02377-f007:**
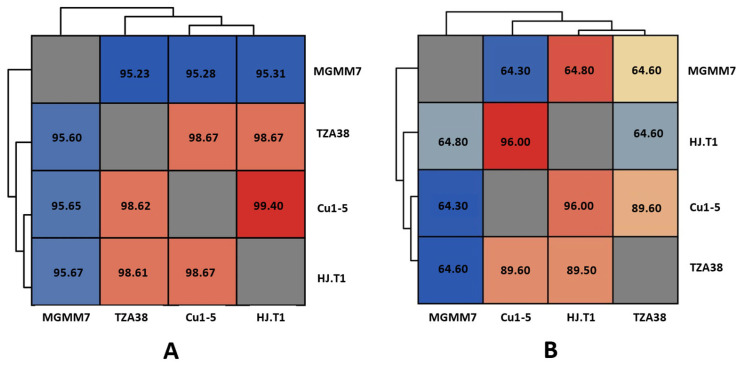
Average nucleotide identity based on BLAST (ANIb) (**A**) and digital DNA–DNA hybridization (dDDH) (**B**) estimation among *L. fusiformis* strains using JSpeciesWS and Genome-to-Genome Distance Calculator (GGDC) (formula 2 used was HSP length/total length). A distance metric (Euclidean distance) was used to determine the relatedness between strains. The resulting distances (%) were organized into an ANI and dDDH matrix, clustered based on distance patterns, and visualized as a color-coded heatmap. The heat map was generated using ClusVis.

**Figure 8 microorganisms-12-02377-f008:**
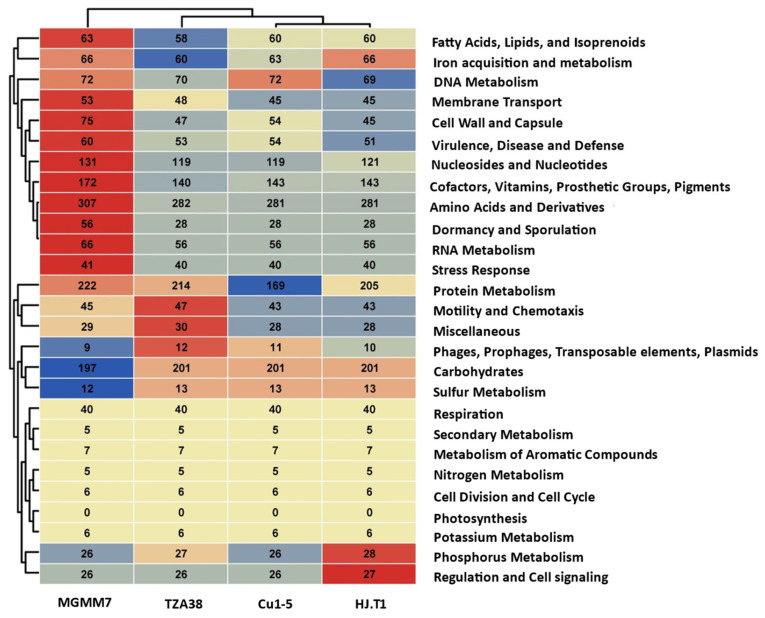
Comparison of subsystem distribution among different categories based on RAST SEED. Heatmap cluster analysis of the subsystem distribution of *L. fusiformis* strains MGMM7, TZA38, Cu1-5, and HJ.T1 using RAST pipeline based on the relative abundances of the non-redundant protein dataset for each genomic strain. The Euclidean distance was used as a metric to evaluate the similarity/dissimilarity between each pair of abundance strain profiles.

**Figure 9 microorganisms-12-02377-f009:**
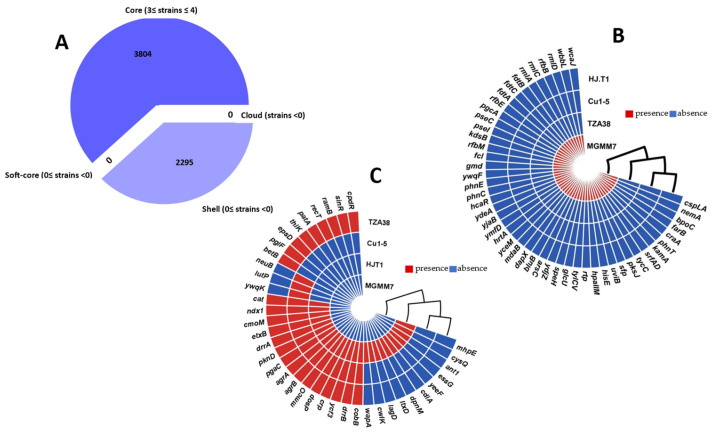
Pangenome analysis of *L. fusiformis* strains. (**A**) Pie chart of the pangenome showing the core, soft core, shell, and cloud genes. (**B**) Heatmap showing the genes expressed in *L. fusiformis* strains TZA38, HJ.T1, and Cu1-5 and absent in *L. fusiformis* MGMM7. (**C**) Heatmap showing genes expressed in *L. fusiformis* MGMM7 but absent in TZA38, Cu1-5, and HJ.T1. *L. fusiformis* strains are listed following hierarchical clustering created using a Manhattan distance matrix based on the gene presence/absence gene content matrix.

**Figure 10 microorganisms-12-02377-f010:**
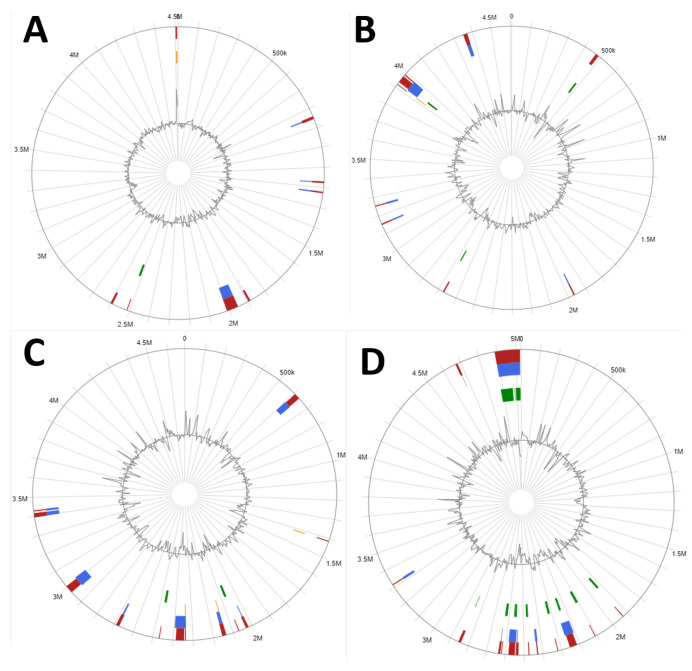
A circular plot of genomic island distribution in *L. fusiformis* strain Cu1-5 (**A**), TZA38 (**B**), HJ.T1 (**C**), and MGMM7 (**D**) was predicted using genomic IslandViewer 4. Red, blue, and green bars represent the genetic elements predicted in each genome using the integrated, IslandPath-DIMOB, and IslandPick methods, respectively.

**Table 1 microorganisms-12-02377-t001:** Experimental design used in this study.

Group	Composition
1	Untreated soil (garden soil)
2	Soil amended with crude oil at a ratio of 1:1 (m/m)
3	Soil amended with crude oil at a ratio of 1:1 (m/m) and MGMM7 suspension at a concentration of 50 mL/kg of soil.

**Table 2 microorganisms-12-02377-t002:** Experimental design used to evaluate azo dye degradation ability of *L. fusiformis* MGMM7.

Experiment	Incubation Conditions at 30 ± 1 °C
No. 1	Static condition for 24 h
No. 2	Shaking condition at 150 rpm for 24 h
No. 3	Static condition for 24 h then with agitation at 150 rpm for 24 h
No. 4	Shaking condition at 150 rpm for 72 h
No. 5	Static condition for 72 h

**Table 3 microorganisms-12-02377-t003:** *Lysinibacillus fusiformis* genomic strains analyzed in this study.

Bacterial Strains	Isolation Source	GenBank Accession	Reference
*L. fusiformis* MGMM7	Wheat rhizosphere	CP130331.1	In this study
*L. fusiformis* TZA38	Sinkhole	CP141829.1	-
*L. fusiformis* HJ.T1	Polyester fabric recovered from compost	CP104728.1	-
*L. fusiformis* Cu1-5	Sludge	CP031773.1	[38]

**Table 4 microorganisms-12-02377-t004:** General genome features of *L. fusiformis* strains analyzed in this study.

Genome Characteristics		Cu1-5	HJ.T1	TZA38	MGMM7
Chromosome	Size (bp)	4,514,433	4,685,025	4,607,073	5,028,939
	GC (%)	37.5	37.5	37.5	37.34
	Genes	4406	4712	4446	5105
	rRNA	80	149	150	162
	tRNA	68	107	108	119
	Pseudogenes	86 (1.95%)	53 (1.12%)	46 (1.03%)	68 (1.33%)
Plasmid	Size (bp)	-	171,063	-	244,822
	GC (%)	-	33.5	-	36.92
	Genes	-	198	-	249

## Data Availability

The original contributions presented in the study are included in the article/Appendix A, further inquiries can be directed to the corresponding authors.

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
