# Peer review of "Genomic Insights of Wheat Root-Associated *Lysinibacillus fusiformis* Reveal Its Related Functional Traits for Bioremediation of Soil Contaminated with Petroleum Products"

_microorganisms, 2024, doi:10.3390/microorganisms12112377_

Round 1

Reviewer 1 Report

Comments and Suggestions for Authors

The current examines the biodegradation potential of an environmental isolate of L. fusiformis, on several different pollutants including crude oil, azo dye and phenol. The originality of the presented work is high, and therefore, it is of interest to the readership of the journal. It also has conducted a comprehensive study on L. fusiformis biodegradation potential of many important pollutants. However, prior to publication, please consider the following points;

1. No or very little information on the biodegradation/biotransformation  products of all pollutants are given in this paper. This is a major flaw of the current version of the manuscript. For example, phenol degradation may yield catechol derivatives, as a result of aerobic degradation activity catalyzed by mono/di oxygenase enzymes. There's no such experimental evidence presented, pertaining to the over expression of such genes either by qPCR or by actual chromatographic identification/quantification of such intermediates.

2. Same can be said about azo dye degradation. The general consensus on bacterial azo dye degradation is that they first undergo anaerobic cleavage of the azo linkages (decolorization), resulting in the formation of aromatic amines, followed by aerobic aromatic ring opening reactions by oxygenase enzymes. No experimental evidence or any discussion is provided in this paper for such degradation mechanisms in this paper. This is one of the major flaws of this study and it needs to be adequately addressed in revisions. 

Author Response

Dear Reviewer,

We are grateful for your consideration of this manuscript and thanks for your careful reading of our text. We appreciate your suggestions, which have been very helpful in improving the manuscript. All the comments that we received on this study have been taken into account and we present our reply to each of them separately.

Kindly find below our response to your comments.  All changes in the initial version of the manuscript and figures are in blue font color for added sentences and strikethrough in red font color for deleted words or sentences.

Two versions of the manuscript are enclosed, one where all the changes have been underlined, and another version without any marks. We hope that these changes to the manuscript will facilitate the decision to publish this study in your journal. We have made a considerable effort to take into account the interesting suggestions proposed by the reviewers. In any case, we are open to consideration of any further comments on our answers.

Authors’ response to comments and suggestions

Below we respond to the Reviewers’ comments.

(Note: Reviewers’ comments are in italic, and authors' responses are in bold).

Reviewer 1

The current examines the biodegradation potential of an environmental isolate of L. fusiformis, on several different pollutants including crude oil, azo dye and phenol. The originality of the presented work is high, and therefore, it is of interest to the readership of the journal. It also has conducted a comprehensive study on L. fusiformis biodegradation potential of many important pollutants. However, prior to publication, please consider the following points.

  • No or very little information on the biodegradation/biotransformation products of all pollutants is given in this paper. This is a major flaw of the current version of the manuscript. For example, phenol degradation may yield catechol derivatives as a result of aerobic degradation activity catalyzed by mono/dioxygenase enzymes. There's no such experimental evidence presented about the over-expression of such genes either by qPCR or by actual chromatographic identification/quantification of such intermediates.

Thank you for your observation. In this work, we focused on the genomic structural analysis of Lysinibacillus fusiformis and a qualitative assessment of its xenobiotic degradation capabilities. We did not delve into a detailed investigation of the specific pathways and degradation products of petroleum, phenol, and azo dyes. We recognize the gap in our manuscript concerning the products generated by plant-associated L. fusiformis after phenol or azo dye degradation. These experiments will be included in our next research paper. However, to address the current gap, we have incorporated recent literature data into the Discussion section, providing insights into potential pathways and products which may be associated with azo dye and phenol degradation of L. fusiformis.

  • Same can be said about azo dye degradation. The general consensus on bacterial azo dye degradation is that they first undergo anaerobic cleavage of the azo linkages (decolorization), resulting in the formation of aromatic amines, followed by aerobic aromatic ring opening reactions by oxygenase enzymes. No experimental evidence or any discussion is provided in this paper for such degradation mechanisms in this paper. This is one of the major flaws of this study and it needs to be adequately addressed in revisions.

We agree with the reviewer that the anaerobic degradation of azo dyes is significant due to the oxygen sensitivity of azo reductase enzymes. However, it's important to note that aerobic degradation can also be effective, as demonstrated by various studies, including Sari & Simarani (2019). In their study, Lysinibacillus fusiformis was able to decolourize methyl red under both static and shaking conditions with similar efficiency. To further investigate the impact of different cultivation conditions on azo dye degradation by L. fusiformis MGMM7, we conducted additional experiments comparing static, static + shaking, and shaking conditions. These results, along with a more detailed discussion of the various mechanisms involved in xenobiotic degradation, have been incorporated into the revised manuscript. By considering both aerobic and anaerobic conditions, we aim to gain a comprehensive understanding of the factors influencing azo dye degradation by L. fusiformis and to optimize bioremediation strategies.

Reference

Sari, I. P., & Simarani, K. (2019). Comparative static and shaking culture of metabolite derived from methyl red degradation by Lysinibacillus fusiformis strain W1B6. Royal Society Open

Reviewer 2 Report

Comments and Suggestions for Authors

The paper examines a Lysinibacillus fusiformis strain isolated from wheat root for its bioremediation potential in terms of petroleum oil, phenol and azo dye degradation, as well as the analysis of its genome responsible for these properties. The topic is interesting and attractive, with a well structured and easy to read and understandable manuscript. The sections of the manuscript are in the right order, where all of the discussion supports the presented results. There is sufficient comparison of obtained results with previously published data. I can gladly recommend it to be published after minor issues have been addressed:

1. In the last few sentences of the Introduction section please emphasize the novelty of the work, i.e. is the isolated strain a new one and weather this gene sequencing has been done before or not.

2. In section 2.3. the analysis is only for the determining the presence and the quantity of produced biosurfactants?

3. In section 2.6. please describe how was the germination rate determined.

4. In Figure 4 it can clearly be seen that for the BM medium the biomass reached stationary phase of growth after 38 h, and for the BM + Phenol it also reached this point and then increased its growth for the second time. Could it be that the strain first utilized what it could from BM and them from 38 h switched to phenols? It would be interesting to examine further, maybe the biosurfactants produced in these two parts of cultivation are different…

5. In the initial analysis the supernatant showed identical results as the cell suspension, hence the conclusion that the produced biosurfactants are responsible for the degradation of oil. Why hasn’t this comparative analysis (both cells suspension and supernatant) continued in the following experiments?

Finally, I need to say that I am eager to see what future work on this topic will bring, especially in terms of examining, optimizing and scaling-up the production the cells suspension or supernatant (biosurfactants). Good luck!

Author Response

Dear Reviewer,

We are grateful for your consideration of this manuscript and thanks for your careful reading of our text. We appreciate your suggestions, which have been very helpful in improving the manuscript. All the comments that we received on this study have been taken into account and we present our reply to each of them separately.

Kindly find below our response to your comments.  All changes in the initial version of the manuscript and figures are in blue green font color for added sentences and strikethrough in red font color for deleted words or sentences.

Two versions of the manuscript are enclosed, one where all the changes have been underlined, and another version without any marks. We hope that these changes to the manuscript will facilitate the decision to publish this study in your journal. We have made a considerable effort to take into account the interesting suggestions proposed by the reviewers. In any case, we are open to consideration of any further comments on our answers.

Authors’ response to comments and suggestions

Below we respond to the Reviewers’ comments.

(Note: Reviewers’ comments are normal, and authors' responses are in bold).

Reviewer

The paper examines a Lysinibacillus fusiformis strain isolated from wheat root for its bioremediation potential in terms of petroleum oil, phenol and azo dye degradation, as well as the analysis of its genome responsible for these properties. The topic is interesting and attractive, with a well-structured and easy to read and understandable manuscript. The sections of the manuscript are in the right order, where all of the discussion supports the presented results. There is sufficient comparison of obtained results with previously published data. I can gladly recommend it to be published after minor issues have been addressed:

  1. In the last few sentences of the Introduction section please emphasize the novelty of the work, i.e. is the isolated strain a new one and whether this gene sequencing has been done before or not.

We thank the reviewer for pointing out this issue. Appropriate changes have been made.

Original text (Lines 88-93): This study evaluated the ability of the plant-associated bacterium L. fusiformis MGMM7 isolated from the rhizosphere of winter wheat (Triticum aestivum L.) to bioremediate and biodegrade environmental pollutants, such as azo dyes, crude oil, and by-products. Additionally, we also sequenced the whole genome of MGMM7 and compared it to related genomes of L. fusiformis strains, specifically those isolated from sources containing xenobiotic contaminants.

Modified text: As stated above, for green remediation primarily based on active microbial metabolites, microbes are typically from relevant sources, since they possess metabolic pathways and enzyme systems that enable them to degrade specific pollutants more efficiently. Lysinibacillus fusiformis strains used as biodegradation agents for crude oil reported in recent literature have been isolated from sources contaminated with xenobiotic pollutants (Wemedo et al., 2018; Ifeoluwa et al., 2020; John et al.,2021 ). Few studies have evaluated whether strains of this species isolated from uncontaminated or nutrient-rich sources such as the rhizosphere possess similar bioremediation properties. This study assessed the ability of the plant-associated bacterium L. fusiformis MGMM7 isolated from the rhizosphere of winter wheat (Triticum aestivum L.) to bioremediate and biodegrade environmental pollutants, such as azo dyes, crude oil, and byproducts. Additionally, we also sequenced the whole genome of MGMM7 and compared it to related genomes of L. fusiformis strains, specifically those isolated from sources containing xenobiotic contaminants.

Reference

John, W. C., Ogbonna, I. O., Gberikon, G. M., & Iheukwumere, C. C. (2021). Evaluation of biosurfactant production potential of Lysinibacillus fusiformis MK559526 isolated from automobile-mechanic-workshop soil. Brazilian Journal of Microbiology, 52, 663-674.

Wemedo, S. A., Nrior, R. R., & Ike, A. A. (2018). Biodegradation potential of bacteria isolated from crude oil polluted site in South-South, Nigeria. Journal of Advances in Microbiology, 12(2), 1-13.

Ifeoluwa, S. E., Theophilus, T. E., Ogunware, A. E., Oyende, Y. E., & Onakomaiya, A. O. (2020). Degradation of phenanthrene and some selected petroleum hydrocarbons by Lysinibacillus fusiformis (ALSL 5). J Exp Sci, 11, 5-10.

  1. In section 2.3. The analysis is only for determining the presence and the quantity of produced biosurfactants?

In section 2.3, we focused on a qualitative analysis to determine the presence or absence of biosurfactants in the cell-free supernatant. While this provides valuable information about the production potential of the strain, it doesn't quantify the exact amount of biosurfactant produced.

  1. In section 2.6. Please describe how was the germination rate determined.

We agree with your keen observation and method used to evaluate the germination rate was added in the initial version of the manuscript (Lines 160– 164).

Original text (Lines 160–164): The germination rate was analyzed throughout the experiment.

Modified text: The germination rate was analyzed as the proportion of sprouted seeds (PSS), expressed as a percentage of the total number of seeds (TNS) taken for germination as described in following equation :

Thanks again for your insight and comments that helped improve our manuscript.

  1. In Figure 4 it can clearly be seen that for the BM medium the biomass reached the stationary phase of growth after 38 h, and for the BM + Phenol it also reached this point and then increased its growth for the second time. Could it be that the strain first utilized what it could from BM and them from 38 h switched to phenols? It would be interesting to examine further, maybe the biosurfactants produced in these two parts of cultivation are different…

We agree with the reviewer’s comment. As noted in the discussion section (lines 609-612), MGMM7 initially prioritizes glucose as its preferred carbon source. It efficiently uses glucose when other carbon sources are available, followed by the sequential utilization of other carbon sources. Thank you for your careful observation. We've also corrected a minor error regarding the secondary exponential growth phase in BM + Phenol, which occurred after 38 hours instead of 34 hours. Generally, bacteria grown in a minimal medium supplemented with both glucose and phenol will produce more biosurfactants. Phenol acts as a stressor for bacteria, prompting them to upregulate biosurfactant production as a survival mechanism (Reddy et al., 2018). Additionally, biosurfactants help bacteria utilize phenol as a carbon source by increasing its bioavailability. By increasing the bioavailability of phenol, biosurfactants enable bacteria to more efficiently transport and metabolize this compound as a carbon source. In our future research, we plan to optimize biosurfactant production and evaluate its effectiveness in emulsification and micelle formation. These properties are crucial for bioremediation of environments contaminated with hydrophobic pollutants, where biosurfactant-producing bacteria play a key role.

Reference

Reddy, P. V., Karegoudar, T. B., & Nayak, A. S. (2018). Enhanced utilization of fluorene by Paenibacillus sp. PRNK-6: Effect of rhamnolipid biosurfactant and synthetic surfactants. Ecotoxicology and Environmental Safety, 151, 206-211.

  1. In the initial analysis the supernatant showed identical results as the cell suspension, hence the conclusion that the produced biosurfactants are responsible for the degradation of oil. Why hasn’t this comparative analysis (both cell suspension and supernatant) continued in the following experiments?

Biosurfactants play a crucial role in the biodegradation of crude oil. They increase bioavailability by reducing surface tension between oil and water, breaking down large oil droplets into smaller ones, making them more accessible to oil-degrading microorganisms (Santoso et al., 2016; Patowary et al., 2017). Biosurfactants can also enhance microbial adhesion, improving the adhesion of microorganisms to oil droplets, facilitating direct contact and further enhancing the biodegradation process (Nikolova & Gutierrez 2021). In our case, we checked for the presence of biosurfactants in the cell-free supernatant to ensure the ability of our isolated strain to produce these amphiphilic compounds. Therefore, we concluded that there was no need to compare the oil degradation abilities of cell suspensions and cell-free supernatants.

Reference

Patowary, K., Patowary, R., Kalita, M. C., & Deka, S. (2017). Characterization of biosurfactant produced during degradation of hydrocarbons using crude oil as sole source of carbon. Frontiers in microbiology, 8, 279.

Santos, D. K. F., Rufino, R. D., Luna, J. M., Santos, V. A., & Sarubbo, L. A. (2016). Biosurfactants: multifunctional biomolecules of the 21st century. International journal of molecular sciences, 17(3), 401.

Nikolova, C., & Gutierrez, T. (2021). Biosurfactants and their applications in the oil and gas industry: current state of knowledge and future perspectives. Frontiers in Bioengineering and Biotechnology9, 626639.

  1. Finally, I need to say that I am eager to see what future work on this topic will bring, especially in terms of examining, optimizing and scaling-up the production the cells suspension or supernatant (biosurfactants). Good luck!

Thank you for your kind words and valuable feedback. We are equally excited about the future potential of this research. Your suggestion to optimize and scale up the production of biosurfactants is insightful and will be a key focus in our future work. We believe that by further developing these technologies, we can make significant contributions to environmental bioremediation.

Round 2

Reviewer 1 Report

Comments and Suggestions for Authors

All concerns raised during peer review has been adequately addressed in the revised version. This paper can be accepted for publication